# Rapid, Single-Step Protein Encapsulation via Flash NanoPrecipitation

**DOI:** 10.3390/polym11091406

**Published:** 2019-08-27

**Authors:** Shani L. Levit, Rebecca C. Walker, Christina Tang

**Affiliations:** Chemical and Life Science Engineering Department, Virginia Commonwealth University, Richmond, VA 23284-3028, USA

**Keywords:** Flash NanoPrecipitation, nanoparticles, polyethylenimine, self-assembly, tannic-acid, electrostatic interactions, protein encapsulation

## Abstract

Flash NanoPrecipitation (FNP) is a rapid method for encapsulating hydrophobic materials in polymer nanoparticles with high loading capacity. Encapsulating biologics such as proteins remains a challenge due to their low hydrophobicity (logP < 6) and current methods require multiple processing steps. In this work, we report rapid, single-step protein encapsulation via FNP using bovine serum albumin (BSA) as a model protein. Nanoparticle formation involves complexation and precipitation of protein with tannic acid and stabilization with a cationic polyelectrolyte. Nanoparticle self-assembly is driven by hydrogen bonding and electrostatic interactions. Using this approach, high encapsulation efficiency (up to ~80%) of protein can be achieved. The resulting nanoparticles are stable at physiological pH and ionic strength. Overall, FNP is a rapid, efficient platform for encapsulating proteins for various applications.

## 1. Introduction

Flash NanoPrecipitation (FNP) is a versatile method to incorporate hydrophobic drugs, dyes, or inorganic nanoparticles with a hydrophobic coating into polymeric nanoparticles via rapid mixing achieved with confined impinging jets [1]. Typically, nanoparticle self-assembly involves precipitation of the supersaturated hydrophobic material via nucleation and growth as well as adsorption of a micellizing amphiphilic block copolymer. The resulting nanoparticles are sterically stabilized with a hydrophobic core—hydrophilic shell structure [1,2]. Due to the rapid precipitation rate and strong hydrophobic interaction with the hydrophobic block of the amphiphilic block copolymer necessary for stabilization, the use of FNP has generally been limited to encapsulation of hydrophobic materials (logP > 6) [3]. Due to the increasing emphasis on biologically derived therapeutics [4] such as proteins and peptides, encapsulation to prevent rapid clearance from natural mechanisms and enzymatic degradation of the biologics [5,6] via FNP is of considerable interest. 

Encapsulation of less hydrophobic materials (logP < 6) [7,8,9,10] using FNP has been achieved via *in situ* complexation [7,8,9,10]. For example, hydrophobic ion pairs [9] or insoluble coordination complexes can be formed during mixing and stabilized with an amphiphilic block copolymer [10]. To encapsulate peptides, hydrophilic imaging agents, and small proteins (~14 kDa), inverse Flash NanoPrecipitation (iFNP) has recently been reported [11,12]. In iFNP, the biologic and the amphiphilic block copolymer are solubilized in a polar organic solvent (e.g., dimethyl sulfoxide) and rapidly mixed with a miscible nonpolar solvent (e.g., acetone or chloroform) which leads to precipitation of the biologic, adsorption of the hydrophilic block, and stabilization by the hydrophobic block in the nonpolar solvent. These initial particles, with a hydrophilic core and hydrophobic coating, are then crosslinked for stabilization and dispersed in an appropriate solvent for a second FNP step for encapsulation within a second block copolymer [11,12]. While promising, this approach inherently requires multiple processing steps. 

Another approach to encapsulate biologics has been Flash Nanocomplexation in which polyelectrolytes complex with biologics (e.g., negatively charged DNA) to impart stability. For gene delivery, Santos et al. stabilized DNA with linear polyethylenimine (*l*PEI) (22 kDa) via rapid mixing [13]. This approach of leveraging electrostatic interactions has been successful for strong polyelectrolytes such as DNA. Use of this approach for encapsulation of diffusely charged, globular proteins has not yet been established. 

Therefore, we investigate a single-step method for encapsulation of proteins via FNP. We use bovine serum albumin (BSA) as a model protein and form an insoluble precipitate *in situ* with tannic acid [14,15,16]. We study various stabilizers (i.e., an amphiphilic block copolymer and a polyelectrolyte). The effects of formulation parameters e.g., stabilizer concentration, molecular weight, pH, and ionic strength on nanoparticle size, zeta potential, stability, protein encapsulation efficiency are discussed.

## 2. Materials and Methods

### 2.1. Materials

ACS grade tannic acid (TA), calcium chloride (CaCl_2_), and ACS grade hydrochloric acid (HCl) were purchased from Sigma-Aldrich (St. Louis, MO, USA). The branched polyethylenimine (PEI) with weight average molecular weight (M_W_) of 2000 g/mol and 10,000 g/mol, were obtained from PolySciences (Warrington, PA, USA), and M_W_ = 750,000 g/mol 50% (w/v) in H_2_O was obtained from Sigma-Aldrich (St. Louis, MO, USA). Bovine serum albumin (BSA), ACS grade acetone, HPLC grade tetrahydrofuran (THF), diethyl ether, ammonium hydroxide (NH_4_OH) (aq. 10% v/v), and ACS certified sodium chloride (NaCl) were obtained from Fisher Scientific (Pittsburg, PA, USA). These reagents were used as received. Phosphate-buffered saline (PBS) was made with 156 mM NaCl, 10 mM sodium phosphate dibasic anhydrous (Na_2_HPO_4_, Fisher Scientific, Pittsburg, PA, USA), and 2 mM potassium phosphate monobasic (KH_2_PO_4_, Fisher Scientific, Pittsburg, PA, USA). Prior to use, BioRad Protein Assay Dye Reagent (Bradford dye, BioRad, Hercules, CA, USA) was filtered with a 0.45 µm PTFE filter (Fisher Scientific, Pittsburg, PA, USA) and diluted 4-fold with deionized water. Additionally, polystyrene-b-polyethylene glycol (1600-b-500 g/mol) (PS-b-PEG) obtained from Polymer Source (Product No. P13141-SEO, Montreal, Canada) was dissolved in THF (500 mg/mL) and precipitated in diethyl ether (1:20 v:v THF:ether). The PS-b-PEG was recovered by centrifuging, decanting, and drying under vacuum at room temperature for two days as previously described [17].

### 2.2. Nanoparticle Preparation

Flash NanoPrecipitation (FNP) was performed with a hand-operated confined impinging jet (CIJ) mixer similar to previous reports [10]. Using the amphiphilic block copolymer stabilizer, PS-b-PEG was dissolved with TA (5 mg/mL) in acetone by sonication (~40 °C) and rapidly mixed with BSA dispersed in water (9 mg/mL or 20 mg/mL). The block copolymer to core material (BSA/TA) ratio was 2:1 by mass. The effluent from the CIJ mixer was immediately diluted in deionized water to maintain an acetone:water volume ratio of 1:9.

To use PEI as a stabilizer, TA (5 mg/mL) was dissolved in acetone and rapidly mixed with BSA dispersed in water (9 mg/mL); the mixer effluent was immediately diluted into PEI dispersed in water. The amount of PEI was varied relative to the mass of BSA/TA to achieve a final ratio PEI:BSA/TA of 1:1, 2:1, or 3:1 by mass. The volume of aqueous PEI was set to maintain a final acetone/water ratio of 1:9 by volume. To determine the role of TA, FNP was performed without TA using PEI as a stabilizer by rapidly mixing BSA dispersed in water with acetone and immediately diluting with PEI dispersed in water. In some cases, the ionic strength or the pH of the aqueous BSA stream was adjusted with HCl and NH_4_OH to achieve pH values between 2 and 10. In some cases following FNP, dialysis was performed to remove the organic solvent using regenerated cellulose tubing with a molecular weight cutoff of 6–8 kD MWCO (Spectra/Por, Spectrum Laboratories, Houston, TX, USA) against deionized water at a ratio of 1:100. The bath water was replaced four times in a 24-hour period.

### 2.3. Nanoparticle Characterization

The size and zeta potential of the resulting nanoparticle dispersions were measured after formulation with a Malvern Zetasizer ZS with a backscatter detection angle of 173° (Malvern Instruments Ltd., Malvern, United Kingdom). The intensity average particle size and distribution are reported using normal resolution mode with an average of 4 measurements. The polydispersity index (PDI) is used as a measure of the breadth of the particle distribution defined from the moments of the cumulant fit of the autocorrelation function calculated by the instrument software as previously described. Nanoparticles with a PDI below 0.300 were considered uniform [10,18].

To assess nanoparticle stability, the pH of the nanoparticle dispersions following FNP was adjusted between pH 2 and 10 with HCl or NH_4_OH. The size and zeta potential were tracked by DLS for up to a week. Additionally, the effect of ionic strength on particle stability by adding NaCl or CaCl_2_ (10 to 300 mM) after FNP and the resulting size and zeta potential were tracked for 24 h.

### 2.4. Protein Quantification

The amount of protein encapsulated in the resulting nanoparticles was quantified using a Bradford assay. After FNP, nanoparticles were recovered by centrifugal filtration (Amicon Ultracel 50K, 50,000 NMWL, Merck Millipore Ltd., Burlington, MA, USA). Briefly, filters were centrifuged (5804 R 15 amp. version, Eppendorf, Hamburg, Germany) at ~4000 rpm for 30–40 min. The recovered nanoparticles were separated from the supernatant. The recovered particles were washed 3 times with acetone (1 mL) to precipitate the BSA and solubilize the other nanoparticle components. The precipitated BSA was recovered from the acetone soluble nanoparticle components by centrifugation (10,000 rpm for 5–10 min) and decanting the acetone supernatant. The recovered protein was redispersed in water. A Bradford assay was performed on the sample following the manufacture’s protocol. Briefly, 10 µL of sample and 200 µL of Bradford dye were added to 96-well plate and measured with a microplate reader (VersaMax ELISA microplate reader, Molecular Devices, San Jose, CA, USA or Cytation 3 multi-mode reader, BioTek, Winooski, VT, USA) at a wavelength of 595 nm. Performing the procedure with a known amount of BSA, we confirmed 98 ± 3% protein recovery (Appendix A).

## 3. Results and Discussion

Tannins such as tannic acid (TA) are known to precipitate proteins. Thus, to encapsulate proteins via Flash NanoPrecipitation (FNP), our approach was to form an insoluble complex with tannins during mixing in the presence of a stabilizer to facilitate nanoparticle self-assembly and impart stability. We use TA and bovine serum albumin (BSA) as a model system. Initially, we examined the precipitation of the BSA-TA complex via FNP. We mixed BSA dispersed in water with TA dissolved in acetone, which immediately formed a cloudy dispersion with a zeta potential of −13.1 ± 0.6 mV (Table 1). Without a stabilizer, the precipitate continued to grow and macroscopic precipitation was observed within 24 h. These observations indicate that BSA and TA complex and precipitate sufficiently fast upon mixing for nanoparticle self-assembly with FNP. We varied the ratio of BSA to TA (between 3:7 and 7:3 by mass) and observed the amount of macroscopic precipitate that formed. A mass ratio of 9:5 BSA to TA, produced the greatest amount of macroscopic precipitate (Appendix A), and was thus used for subsequent experiments.

Based on these results of rapidly precipitating BSA with TA, we initially formulated nanoparticles using an amphiphilic block copolymer (PS-b-PEG) as a stabilizer. To perform FNP, BSA was dispersed in water and rapidly mixed with TA and PS-b-PEG which were dissolved in acetone. At a BSA to TA mass ratio of 9:5 and a block copolymer to core mass ratio of 2:1, the nanoparticle dispersion was polydisperse with multiple peaks at ~100 nm, ~20 nm, and ~10 nm (Appendix A). The peaks can be attributed to TA/PS-b-PEG micelles [10], empty PS-b-PEG micelles [19], and soluble BSA [20], respectively. The lack of visible TA/BSA precipitate suggests that TA preferentially interacts with the block copolymer rather than with BSA during FNP. Further, there is a mismatch in timescales of complexation/precipitation and block copolymer micellization such that the block copolymer rapidly forms micelles and on a longer time scale stabilizes TA [10,21].

To promote BSA/TA interactions, we performed FNP with an excess of protein. When the BSA to TA ratio was increased to 4:1, nanoparticles were initially formed with a size of ~600 nm with a PDI of 0.347 ± 0.045 similar to BSA-TA without stabilizer (Figure 1A,B). The measured zeta potential, −18.0 ± 3.0 mV, is consistent with other PEG based block copolymer stabilized nanoparticles [3,22]. Therefore, it appears that upon mixing BSA and TA complex and precipitate then the hydrophobic block of the amphiphilic block copolymer stabilizes the precipitate. TA also undergoes intermolecular interactions with the PEG block of the block copolymer via hydrogen bonding forming an insoluble complex that is confined to the nanoparticle core with the TA/BSA precipitate. The hydrophobic block of the block copolymer adsorbs to the precipitating nanoparticle core (hydrophobic PS-block, TA/BSA precipitate, TA:PEG) due to hydrophobic interactions. The PEG that is not complexed with TA microphase separates from the PS-block and orients into the aqueous phase providing steric stabilization. In this case, nanoparticle assembly is driven by a combination of hydrophobic and hydrogen bonding interactions.

While PS-b-PEG initially facilitated nanoparticle self-assembly, the resulting nanoparticle dispersion, initially transparent, turned cloudy over several hours indicating nanoparticle growth. Over 24 h, TA and BSA partitioned out and re-precipitated outside of the nanoparticle core. Similar behavior has been observed with TA [10] and peptides [11] which is attributed to low affinity between the hydrophobic block of PS-b-PEG and the BSA/TA precipitate. 

Therefore, we next considered alternative stabilizers. Since we observed the initial BSA/TA complex showed a negative zeta potential of −13.1 ± 0.6 mV, we considered a cationic polyelectrolyte, polyethylenimine (PEI). To perform FNP, BSA dispersed in water was rapidly mixed with TA dissolved in acetone. The effluent of the mixer was immediately diluted into PEI with a molecular weight of 750,000 g/mol (750 kDa PEI) dispersed in water. The resulting nanoparticles were 107 ± 5 nm with a PDI 0.285 ± 0.004 (Table 2). TEM imaging confirms that the particles are spherical and the size is consistent with DLS measurements (Appendix A). No macroscopic precipitate was observed over at least 7 days whereas macroscopic precipitate was observed within 24 h without a stabilizer (Figure 1C). Further, the zeta potential of the resulting +18.8 ± 0.9 mV (Table 2) compared to −13.1 ± 0.6 mV (Table 1) for BSA/TA without a stabilizer. The positive zeta potential suggests that PEI was present at the surface of the nanoparticles encapsulating the anionic BSA-TA precipitate providing some degree of steric stabilization as zeta potentials greater than +35 mV are required for entirely electrostatic stabilization [22]. Taken together, these results indicate that introducing the PEI stabilizer facilitated nanoparticle self-assembly and conferred nanoparticle stability.

Since stabilizer properties can greatly affect the resulting nanoparticle properties [3,23,24,25], we investigated the effect of PEI molecular weight (Appendix A) on nanoparticle assembly and stability. We used molecular weights of 750 kDa, 10 kDa, and 2 kDa. Interestingly, while 750 kDa PEI resulted in 107 ± 5 nm stable nanoparticle, 10 kDa PEI formed monodisperse, stable particles with a diameter of 153 ± 7 nm with a PDI of 0.125 ± 0.022 and a zeta potential +14.4 ± 1.9 mV (Table 2, Appendix A). The 2 kDa PEI did not facilitate nanoparticle assembly (Appendix A) and FNP resulted in macroscopic precipitate. Thus, PEI molecular weights 10 kDa or greater were necessary for nanoparticle formation via self-assembly to encapsulate the BSA/TA complex. High molecular weight polyelectrolytes have been reported to strongly absorb onto surfaces which improve the stabilization of dispersions [26,27] such as the BSA/TA precipitate. In contrast, lower molecular weight polyelectrolytes have higher intermolecular charge repulsion which limits the electrostatic stabilization of the BSA/TA precipitate [27]. 

We confirmed the role of electrostatic interactions in nanoparticle assembly and stabilization by examining the effect of pH of the BSA stream on particle formation. First, we confirmed TA precipitates BSA at various pH conditions; we observed macroscopic precipitation for pH between 7 and 4.5, and no visible precipitate at pH 2 (Appendix A). The maximum amount of visible BSA-TA precipitate was produced around pH 5 which can be attributed to protein aggregation near the isoelectric point of BSA (pI = 4.8) [28] comparable to previous reports [14]. Subsequently, the pH of the BSA stream was adjusted to between 2 and 10 prior to FNP while the PEI reservoir was unbuffered (pH ~ 10) and the nanoparticle size and zeta potential were examined. Varying the pH of the BSA stream did not change the size or zeta potential of the 750 kDa PEI NPs (Appendix A) indicating the measured properties are dictated by the PEI. Interestingly, decreasing the pH to 2 using 10 kDa PEI disrupted particle assembly (Appendix A) and instead formation of a visible precipitate was observed. The net charge of BSA is dependent on pH; decreasing pH below the isoelectric point results in protonation of the protein and a net positive charge [14,16]. With a net positive charge, BSA repels the cationic PEI (pKa ~ 10) [26,29] and thus particles do not form. Therefore, particle assembly requires the pH to be at or above the isoelectric point of the protein (i.e., pH > 5) to ensure electrostatic interaction with the PEI stabilizer. 

Based on these results, we propose that the mechanism of nanoparticle self-assembly differs between the 10 kDa and the 750 kDa PEI. Interestingly, 750 kDa PEI NPs have a size of ~100 nm, similar to the hydrodynamic diameter of 750 kDa PEI (Appendix A). Therefore, the high molecular weight PEI aggregates and these aggregates serve as the nanoparticle template and a sink for absorbing the anionic BSA/TA complex. In contrast, nanoparticles formulated with 10 kDa PEI (~150 nm) are much larger than their corresponding polymer in aqueous media (~5 nm). For this stabilizer, BSA/TA complex and precipitate. Particle assembly occurs as cationic 10 kDa PEI adsorbs to the anionic BSA/TA precipitate via electrostatic interactions. This mechanism of nanoparticle self-assembly is analogous to previous work with FNP encapsulating coordination complexes or ion pairs formed during mixing [7,10,11,12,30]. Schematics of the particle self-assembly mechanisms for the 750 kDa and 10 kDa PEI are shown in Figure 2.

Building on these results, we next sought to understand the effect of formulation parameters on nanoparticle assembly, specifically nanoparticle size. Typically, when nanoparticle assembly occurs due to hydrophobic interactions between the precipitation core material and micellizing block copolymer, the nanoparticle size can be tuned with the mass ratio of the block copolymer to the core materials as well as the total solids concentration [31].

The mass ratio of PEI to BSA-TA complex was adjusted from 3:1 to 2:1. However, for the 750 kDa PEI, decreasing the relative amount of stabilizer resulted in unstable nanoparticles with visible precipitate forming within 24 h (Appendix A). For the 10 kDa PEI, stable particles were achieved with lower relative amounts of stabilizer. While the size was not significantly affected, the PDI increased indicating less uniform particles (Appendix A). Overall, we observe that the range of stabilizer to core ratio that forms stable, uniform nanoparticles is relatively narrow compared to the range used with hydrophobic interactions driving nanoparticle self-assembly. This finding is consistent with particle formation involving *in situ* coordination complexation [10] or cationic polysaccharides [32]. A ratio of 3:1 PEI to BSA-TA complex was used for subsequent experiments.

To vary particle size, the total solids concentration of the BSA, TA, and PEI in the final dispersion was varied from 5.6 mg/mL to 11.2 mg/mL; the 9:5 ratio of BSA:TA and 3:1 ratio of PEI:BSA/TA were held constant. We note that with the 750 kDa PEI stabilizer, the total solids concentration did not affect particle size or stability (Appendix A). Using 750 kDa PEI as a stabilizer, particle assembly was templated by the aggregated polymer [27,29,33]. The results are comparable to previous reports with various PEI systems at constant charge ratios [34,35]. Interestingly, with the 10 kDa PEI stabilizer, doubling the total solids concentration resulted in a two-fold increase in particle size from 143 ± 8 nm to 319 ± 185 nm while maintaining a PDI less than 0.300 (Appendix A). The trend of increasing size with total solids concentration is comparable to previous results with FNP [1,31]. 

Traditionally, FNP involves an amphiphilic block copolymer and hydrophobic core materials. Upon mixing, the rapid decrease in solvent quality leads to simultaneous precipitation of the hydrophobic core material via nucleation and growth and self-assembly of the amphiphilic block copolymer. Nanoparticle assembly is arrested when sufficient hydrophobic block of the amphiphilic block copolymer adsorbs to the precipitating core material preventing further nanoparticle growth and the nanoparticle is sterically stabilized by the water-soluble block of the block copolymer. Typically, the nanoparticle size can be affected by varying the total mass concentration. Specifically, increasing the total mass concentration leads to an increase in particle size which has been attributed to a greater rate of core growth relative to the rate of nucleation which results in larger particle size [1,31]. In this case, the mechanism of particle self-assembly is analogous to traditional FNP because BSA/TA complexation and subsequent precipitation via nucleation and growth is sufficiently fast relative to adsorption of the PEI stabilizer. Thus, these results support the mechanism of particle self-assembly in which TA precipitates the protein and further precipitation is arrested by adsorption of the 10 kDa PEI stabilizer.

Salt and pH are expected to greatly affect electrostatic assemblies [36], thus we examined nanoparticle stability as a function of pH and ionic strength. After mixing, the nanoparticle dispersion had a pH of ~10 due to the PEI. As expected, decreasing the pH to 2 caused the nanoparticles with the 10 kDa PEI stabilizer to disassemble, as indicated by DLS. The presence of a peak on the order of 10 nm can be attributed to unencapsulated BSA [20] (Figure 3A). At acidic conditions, protonation of PEI and BSA leads to a net positive charge on both molecules and charge repulsion, which destabilizes the particles. Similar results were observed after dialysis of the nanoparticles against deionized water (Appendix A) due to decrease in pH near the isoelectric point of BSA. Therefore, the pH of the nanoparticle dispersion should be greater than the isoelectric point of the protein to maintain particle stability. Surprisingly, the particles using the 750 kDa stabilizer were stable below the isoelectric point of BSA (pH < 4.8) when both the PEI and BSA are expected to carry a net positive charge (Figure 3B). The PEI aggregates may provide a localized buffering effect preventing protonation of BSA [29]. 

To further understand particle stability, we investigated the effect of adjusting the ionic strength of the particle dispersion between 0.01 M to 0.3 M with monovalent and divalent salts. With the 750 kDa PEI stabilizer, the measured size decreased slightly in the presence of salt at ionic strengths greater than 0.01 M (Appendix A). The decrease in particle size with ionic strength has been observed previously and was attributed to decreased intra-molecular charge repulsion forces which allow for tighter PEI aggregate formation [29]. 

In the presence of NaCl, the size of 10 kDa PEI stabilized nanoparticles was not affected at ionic strengths less than 0.03 M and increased two-fold at an ionic strength of 0.3 M. The presence of salts introduce charge screening and reduce the electrostatic interactions between the 10 kDa PEI stabilizer and anionic TA/BSA precipitate leading to the increase in particle size [27,37]. Interestingly, the diameter of the 10 kDa PEI stabilized nanoparticles was 290 ± 6 nm in the presence of NaCl, compared to 188 ± 7 nm with CaCl_2_ at the same ionic strength (0.3 M) (Table 3). The difference in particle size in the presence of Ca^2+^ and Na^+^ ions at the same ionic strength can be attributed to a 3-fold difference in ion concentration of Na^+^ ions compared to Ca^2+^ which resulted in greater charge screening and thus larger particle size. Additionally, calcium promotes protein/TA precipitation compared to sodium [16] and specific BSA-calcium interactions promote BSA aggregation compared to sodium [38], which prevents particle swelling. 

Taken together, these results demonstrate that self-assembled nanoparticles are stable at physiologically relevant ionic strengths (~0.15 M). The particles are pH labile. Specifically, they are stable at basic pH (e.g., 7.4) and disassemble at acidic pH. Such properties may be promising for controlled release application such as intracellular delivery [10,39,40,41]. 

Finally, we quantified the amount of BSA in the nanoparticles in terms of encapsulation efficiency and protein loading for both molecular weights of PEI via Bradford assay. To understand the role of TA in nanoparticle assembly, we compared formulations with and without TA. Interestingly, while TA did not affect particle size or zeta potential (Appendix A), it greatly affected the amount of protein incorporated into the nanoparticles (Table 4). For example, using the 750 kDa PEI stabilizer, the protein encapsulation increased from 50% with TA to 74% without the presence of TA. This result suggests that the charge density of BSA alone compared to the BSA/TA complex enhances interactions with PEI. Additionally, the absence of TA may increase BSA–PEI interactions. In contrast, with the 10 kDa PEI NPs, the protein encapsulation increased from 8% without TA to 79% with TA. This result supports the mechanism of self-assembly in which TA/BSA complex and precipitate and further growth of the precipitate is prevented by adsorption of the 10 kDa PEI stabilizer.

Interestingly, the 10 kDa PEI stabilizer resulted in higher protein loading (13% compared to 8%) and encapsulation efficiency (79% compared to 50%) than achieved with the 750 kDa PEI stabilizer (Table 4). Nanoparticle assembly using the 10 kDa PEI that occurs due to adsorption of the stabilizer to the precipitate forms kinetically trapped and enhances protein encapsulation compared to the absorption of the precipitate with the 750 kDa PEI aggregates. This improvement in loading is analogous to traditional FNP with hydrophobic small molecules [42]. Excitingly, the encapsulation efficiency of protein via rapid mixing with 10 kDa PEI is greater (~80%) than generally reported for encapsulating biologics via nanoprecipitation (7%–40%) [11,43]. These results suggest that FNP facilitated by TA/BSA complexation and precipitation is a highly efficient, rapid process for encapsulating proteins. Alternatively, we demonstrate it is possible to encapsulate protein via FNP without the need of precipitation using the 750 kDa PEI stabilizer. 

## 4. Conclusions

Overall, we have demonstrated a rapid, single-step method using Flash NanoPrecipitation for encapsulating biologics (i.e., proteins) with high encapsulation efficiency (up to ~80%). Using the 10 kDa PEI stabilizer, nanoparticle formation involves complexation and precipitation with tannic acid and stabilization with a cationic polyelectrolyte. Nanoparticle self-assembly is driven by hydrogen bonding between TA and protein, then electrostatic interactions between the TA/protein precipitate and polyelectrolyte stabilizer. The resulting particles are stable at physiological ionic strengths and pH labile, i.e., stable above the isoelectric point of the protein and disassemble at pH below the isoelectric point of the protein, to facilitate potential controlled release applications.

## Figures and Tables

**Figure 1 polymers-11-01406-f001:**
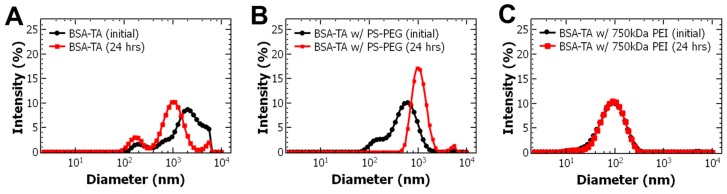
Dynamic light scattering (DLS) intensity weighted size distribution results of (**A**) bovine serum albumin-tannic acid (BSA-TA) complex without the presence of a stabilizer, (**B**) BSA-TA complex with an amphiphilic block copolymer, and (**C**) the BSA-TA complex stabilized with 750kDa polyethylenimine (PEI), immediately upon mixing and after 24 h. The BSA-TA complex stabilized with PEI did not change in size after 24 h.

**Figure 2 polymers-11-01406-f002:**
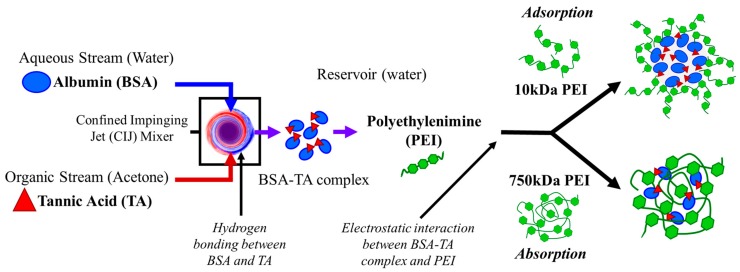
Schematic of the proposed self-assembly mechanisms using 750 kDa and 10 kDa polyethylenimine (PEI) via flash nanoprecipitation (FNP) with PEI stabilizer. In the confined impinging jet (CIJ) mixer the bovine serum albumin (BSA) and tannic acid (TA) interact via hydrogen bonding to form an insoluble complex. Then the complex is immediately diluted in a reservoir containing PEI. The BSA-TA complex interacts with the PEI via electrostatic interaction. High molecular weight 750 kDa PEI aggregates template nanoparticle assembly and absorb the BSA-TA precipitate. In contrast, 10 kDa PEI adsorbs on the precipitating BSA-TA complex forming a core-shell structure.

**Figure 3 polymers-11-01406-f003:**
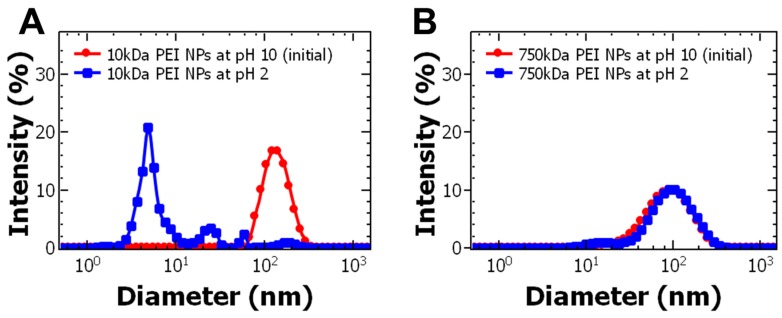
Effect of nanoparticle dispersion pH on size for (**A**) 10 kDa polyethylenimine nanoparticles (PEI NPs) and (**B**) 750 kDa PEI NPs. The size of the particles was measured 24 h after adjusting the pH. The 10 kDa PEI NPs destabilized under acidic conditions and released bovine serum albumin (BSA). The 750 kDa PEI NPs did not change size at acidic pH.

**Table 1 polymers-11-01406-t001:** Zeta potential of BSA-TA complex with polymer stabilizers.

Sample	Zeta Potential (mV)
BSA-TA precipitate	−13.1 ± 0.6
BSA-TA with PS-b-PEG	−18.0 ± 3.0
PEI	+34. 3 ± 4.2
BSA-TA with PEI	+18.8 ± 0.9

**Table 2 polymers-11-01406-t002:** Effect of polyelectrolyte stabilizer molecular weight on nanoparticle properties and stability.

Sample	Initial			7 days		
	Zeta Potential (mV)	Diameter (nm)	PDI	Zeta Potential (mV)	Diameter (nm)	PDI
10 kDa PEI	+15.7 ± 1.0	153 ± 7	0.125 ± 0.022	+14.4 ± 1.9	152 ± 1	0.055 ± 0.013
750 kDa PEI	+18.5 ± 1.3	107 ± 5	0.285 ± 0.004	+18.5 ± 1.3	94 ± 3	0.259 ± 0.011

**Table 3 polymers-11-01406-t003:** Effect of ionic strength on 10 kDa PEI nanoparticle properties.

Salt Added	Concentration (mM)	Ionic Strength (M)	Diameter (nm)	PDI	Zeta Potential (mV)
Initial 10 kDa PEI	0	0	146 ± 2	0.125 ± 0.020	15.7 ± 2.0
NaCl	10	0.01	145 ± 2	0.065 ± 0.019	13.5 ± 2.6
	30	0.03	139 ± 2	0.069 ± 0.007	13.4 ± 2.0
	100	0.1	194 ± 3	0.035 ± 0.023	14.3 ± 0.3
	300	0.3	290 ± 6	0.110 ± 0.024	11.8 ± 1.4
CaCl_2_	10	0.03	138 ± 6	0.351 ± 0.019	15.8 ± 1.0
	100	0.3	188 ± 7	0.148 ± 0.022	16.1 ± 0.5

**Table 4 polymers-11-01406-t004:** Effect of tannic acid (TA) on protein encapsulation.

Sample	Condition	Encapsulation Efficiency (EE%)	Drug Loading (DL%)
10 kDa PEI NPs	no TA	8% ± 3%	1% ± 0%
	with TA	79% ± 7%	13% ± 1%
750 kDa PEI NPs	no TA	74% ± 6%	12% ± 1%
	with TA	50% ± 10%	8% ± 2%

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
