# Peer review of "Rapid, Single-Step Protein Encapsulation via Flash NanoPrecipitation"

_polymers, 2019, doi:10.3390/polym11091406_

Round 1

Reviewer 1 Report

The manuscript by C. Tang et al. reports a rapid, single-step method using Flash NanoPrecipitation for encapsulating biologics with high encapsulation efficiency. Sufficient experiments have been done to demonstrated how it works. The results are interesting and can be accepted for publication in Polymers after revisions.

1. The TEM imaging should be given to show the core-shell structure of PEI coated BSA/TA complex.

2. The authors emphasis “This result has been attributed to a greater rate of core growth relative to the rate of nucleation which results in larger particle size” when talking about the solids concentration effect. But it’s not very clear how the concentration of PEI changes the rate of particle growth and nucleation.

3. The self-assembly mechanisms shown in Figure 2 should also add some detailed information, such as interaction force and the different mechanism between 10 kDa and 750 kDa PEI is confused.

4. Does the size of encapsulated protein have different effect on the final particles?

Author Response

The manuscript by C. Tang et al. reports a rapid, single-step method using Flash NanoPrecipitation for encapsulating biologics with high encapsulation efficiency. Sufficient experiments have been done to demonstrated how it works. The results are interesting and can be accepted for publication in Polymers after revisions.

1. The TEM imaging should be given to show the core-shell structure of PEI coated BSA/TA complex.

We thank the reviewer for their suggestion. We have included TEM of the BSA/TA nanoparticles stabilized with the 750kDa PEI in the supplementary information of the revised manuscript (Figure S3). The images confirm the particles are spherical and the size is consistent with DLS measurements. However, similar to previous reports multiple polymer layers are not visible on the TEM due to low contrast in electron density (Nanomaterials 2019, 9, 318). Our claim that the PEI stabilizes the BSA/TA complex is based on (1) the control experiment in which FNP performed without PEI BSA/TA precipitation is observed and (2) zeta potential measurements which is commonly used to characterize nanoparticles fabricated by layer by layer assembly (e.g. Langmuir 2012, 28, 184-91). In future studies, further structural characterization of the multiple polymer layers could be pursued with Small Angle X-ray Scattering (e.g. ACS Macro Lett. 2017, 69,1005-1012) but is outside the scope of this manuscript focused on the understanding the mechanism of particle assembly. We have revised the discussion to read:

The resulting nanoparticles were 107 ± 5 nm with a PDI 0.285 ± 0.004 (Table 2). TEM imaging confirms that the particles are spherical and the size is consistent with DLS measurements (Figures S3). No macroscopic precipitate was observed over at least 7 days whereas macroscopic precipitate was observed within 24 hours without a stabilizer (Figure 1c). Further, the zeta potential of the resulting +18.8 ± 0.9 mV (Table 2) compared to -13.1 ± 0.6 mV (Table 1) for TA/BSA without a stabilizer. The positive zeta potential suggests that PEI was present at the surface of the nanoparticles encapsulating the anionic BSA-TA precipitate providing some degree of steric stabilization as zeta potentials greater than +35 mV are required for entirely electrostatic stabilization [23]. Taken together, these results indicate that introducing the PEI stabilizer facilitated nanoparticle self-assembly and conferred nanoparticle stability.

2. The authors emphasis “This result has been attributed to a greater rate of core growth relative to the rate of nucleation which results in larger particle size” when talking about the solids concentration effect. But it’s not very clear how the concentration of PEI changes the rate of particle growth and nucleation.

We have clarified this discussion in the revised manuscript.

In traditional FNP, the nanoparticle size can be controlled by varying the concentration ratio of the block copolymer to the core materials. During the mixing, there is a rapid decrease in solvent quality, which precipitates the core materials and simultaneously self-assembles the block copolymer. The absorption of the block copolymer on the particle surface slows the core growth until the nanoparticles are kinetically stable. By varying the ratio of the block copolymer we can increase the rate of core growth relative to the rate polymer adsorption to produce larger nanoparticles. In this section we aim to create a parallel between the self-assembly mechanism of the 10kDa PEI NPs to traditional hydrophobic core nanoparticles formulated with FNP. The revised text:

Interestingly, with the 10kDa PEI stabilizer, doubling the total solids concentration resulted in a two-fold increase in particle size from 143 ± 8 nm to 319 ± 185 nm while maintaining a PDI less than 0.300 (Table S7). The trend of increasing size with total solids concentration is comparable to previous results with FNP. Traditionally FNP involves an amphiphilic block copolymer and hydrophobic core materials. Upon mixing, the rapid decrease in solvent quality leads to simultaneous precipitation of the hydrophobic core material via nucleation and growth and self-assembly of the amphiphilic block copolymer. Nanoparticle assembly is arrested when sufficient hydrophobic block of the amphiphilic block copolymer adsorbs to the precipitating core material preventing further nanoparticle growth and the nanoparticle is sterically stabilized by the water-soluble block of the block copolymer. Typically, the nanoparticle size can be affected by varying the total mass concentration. Specifically, increasing the total mass concentration leads to an increase in particle size which has been attributed to a greater rate of core growth relative to the rate of nucleation which results in larger particle size [1,33]. In this case, the mechanism of particle self-assembly is analogous to traditional FNP because BSA/TA complexation and subsequent precipitation via nucleation and growth is sufficiently fast relative to adsorption of the PEI stabilizer. Thus, these results support the mechanism of particle self-assembly in which TA precipitates the protein and further precipitation is arrested by adsorption of the 10kDa PEI stabilizer.

3. The self-assembly mechanisms shown in Figure 2 should also add some detailed information, such as interaction force and the different mechanism between 10 kDa and 750 kDa PEI is confused.

Based on the reviewer’s suggestions, we have included additional labels of interaction forces during self assembly in Figure 2 and provided more detail in the caption, which now reads:

Figure 2. Schematic of the proposed self-assembly mechanisms using 750kDa and 10kDa polyethylenimine (PEI) via Flash NanoPrecipitation (FNP) with PEI stabilizer. In the confined impinging jet (CIJ) mixer the bovine serum albumin (BSA) and tannic acid (TA) interact via hydrogen bonding to form an insoluble complex. Then the complex is immediately diluted in a reservoir containing PEI. The BSA-TA complex interacts with the PEI via electrostatic interaction. High molecular weight 750kDa PEI aggregates template nanoparticle assembly and absorb the BSA-TA precipitate. In contrast, 10kDa PEI adsorbs on the precipitating BSA-TA complex forming a core-shell structure.

4. Does the size of encapsulated protein have different effect on the final particles?

The reviewer raises an interesting question. In this manuscript, we focus on understanding the mechanism of nanoparticle assembly. Based on the mechanism, we anticipate the approach can be applied to encapsulate other proteins. Previous work suggests that particle size is relatively insensitive to core material molecular weight (for hydrophobic homopolymers) (Nano Lett.2018, 182, 1139-1144) but did not consider in situ complexation. In the future, it would be interesting to study how particle size is affected by protein properties including molecular weight and surface charge distribution (Soft Matter, 2019, 15, 3089-3103) but is outside the scope of this manuscript focused on focused on the understanding the mechanism of particle assembly.

Reviewer 2 Report

In the manuscript “Rapid, Single-Step Protein Encapsulation via Flash NanoPrecipitation” the authors propose a new application of the Flash Nano-Precipitation method with biologics, presenting a protein encapsulation in polymer nanoparticles. They focused on the encapsulation of the model protein BSA with different stabilizers, by characterizing i) the nanoparticles, ii) the effects of formulation parameters on them, and iii) nanoparticle stability as a function of pH and ionic strength.

Interestingly, among the stabilizers tested, polyethylenimine (PEI) has been selected to facilitate nanoparticles self-assembly and stability. PEI has many applications in products like detergents and cosmetics and in biology as a transfection agent. Its use in biology could open the possibility to characterize the nanoparticle assembly proposed, in some cellular systems and to select a class of polymer candidates to be used in NPs formulations.

Moreover, the investigation of the role of TA (tannic acid) in nanoparticle assembly (Table 4 – line 317) it is very interesting not only because of the great protein encapsulation efficiency obtained (80%) as highlighted from the authors. Indeed the increasing in encapsulation efficiency (from 50% with TA to 74% without TA) obtained with 750kDa PEI NPs suggests the possibility to encapsulate proteins in milder conditions, avoiding aggressive protein precipitation that in most cases reduces protein activity and functionality.

Typing corrections:

Figure 3: graph A and graph B are inverted.

Author Response

We thank the reviewer for their insightful comment. We agree and revised the discussion to include:

Excitingly, the encapsulation efficiency of protein via rapid mixing with 10kDa PEI is greater (~ 80%) than generally reported for encapsulating biologics via nanoprecipitation (7 – 40%) [11,45]. These results suggest that FNP facilitated by TA/BSA complexation and precipitation is a highly efficient, rapid process for encapsulating proteins. Alternatively, we demonstrate it is possible to encapsulate protein via FNP without the need of precipitation using the 750kDa PEI stabilizer.

Typing corrections:
Figure 3: graph A and graph B are inverted.

We thank the reviewer for their attention to detail; the revised caption for Figure 3 now reads: Figure 3. Effect of nanoparticle dispersion pH on size for (A) 10kDa polyethylenimine nanoparticles (PEI NPs) and (B) 750kDa PEI NPs. The size of the particles was measured 24 hours after adjusting the pH. The 10kDa PEI NPs destabilized under acidic conditions and released bovine serum albumin (BSA). The 750kDa PEI NPs did not change size at acidic pH.
